# Modulation of Pro-Inflammatory IL-6 Trans-Signaling Axis by Splice Switching Oligonucleotides as a Therapeutic Modality in Inflammation

**DOI:** 10.3390/cells12182285

**Published:** 2023-09-15

**Authors:** Dhanu Gupta, Sara Orehek, Janne Turunen, Liz O’Donovan, Michael J. Gait, Samir El-Andaloussi, Matthew J. A. Wood

**Affiliations:** 1Department of Paediatrics, University of Oxford, Oxford OX3 7TY, UK; 2Biomolecular Medicine, Division of Biomolecular and Cellular Medicine, Department of Laboratory Medicine, Karolinska Institutet, 14151 Huddinge, Sweden; 3Medical Research Council Laboratory of Molecular Biology, Cambridge CB2 0QH, UK

**Keywords:** inflammation, nucleic acid therapies, IL-6 *trans-signaling*, oligonucleotides

## Abstract

Interleukin-6 (IL-6) is a pleiotropic cytokine that plays a crucial role in maintaining normal homeostatic processes under the pathogenesis of various inflammatory and autoimmune diseases. This context-dependent effect from a cytokine is due to two distinctive forms of signaling: *cis*-signaling and *trans-signaling*. IL-6 *cis-signaling* involves binding IL-6 to the membrane-bound IL-6 receptor and Glycoprotein 130 (GP130) signal-transducing subunit. By contrast, in IL-6 *trans-signaling*, complexes of IL-6 and the soluble form of the IL-6 receptor (sIL-6R) signal via membrane-bound GP130. Various strategies have been employed in the past decade to target the pro-inflammatory effect of IL-6 in numerous inflammatory disorders. However, their development has been hindered since these approaches generally target global IL-6 signaling, also affecting the anti-inflammatory effects of IL-6 signaling too. Therefore, novel strategies explicitly targeting the pro-inflammatory IL-6 *trans-signaling* without affecting the IL-6 *cis-signaling* are required and carry immense therapeutic potential. Here, we have developed a novel approach to specifically decoy IL-6-mediated *trans-signaling* by modulating alternative splicing in *GP130*, an IL-6 signal transducer, by employing splice switching oligonucleotides (SSO), to induce the expression of truncated soluble isoforms of the protein GP130. This isoform is devoid of signaling domains but allows for specifically sequestering the IL-6/sIL-6R receptor complex with high affinity in serum and thereby suppressing inflammation. Using the state-of-the-art Pip6a cell-penetrating peptide conjugated to PMO-based SSO targeting GP130 for efficient in vivo delivery, reduced disease phenotypes in two different inflammatory mouse models of systemic and intestinal inflammation were observed. Overall, this novel gene therapy platform holds great potential as a refined therapeutic intervention for chronic inflammatory diseases.

## 1. Introduction

Inflammation is a multifactorial biological process regulating various physiological and pathological processes in response to stimuli, primarily aiming to restore tissue injury and clear pathogens. However, when this process is dysregulated, it can become chronic and potentially induce adverse effects, contributing to the development of autoimmune diseases, such as multiple sclerosis (MS), rheumatoid arthritis (RA), and inflammatory bowel disease [1]. Inflammation is a complex process, which is, to a great extent, orchestrated by various cytokines. Different cytokines can fine-tune inflammatory processes in different ways by inducing and suppressing the inflammatory reactions (i.e., pro-inflammatory or anti-inflammatory, respectively), all eminently dependent upon the type of cytokine involved and the cells influenced by the signaling [1,2]. Recent studies have revealed many roles of cytokines in chronic inflammation, and, among others, interleukin-6 (IL-6) has been profoundly investigated. In the last few decades, IL-6 has been linked to various diseases, such as rheumatoid arthritis (RA), diabetes, cancer and multiple sclerosis (MS) [3], and the overall concentration of IL-6 is elevated in acute and chronic inflammation and is a significant driver of autoimmune diseases [4].

IL-6 is a pleiotropic cytokine with a wide range of functions, from regulating cellular differentiation to proliferation and apoptosis. Intracellular signaling through IL-6 involves two transmembrane receptors. Firstly, IL-6 binds to the IL-6 receptor (IL-6R) with high affinity on the surface of the cell membrane [5]. This complex then further binds to Glycoprotein 130 (GP130 also known as IL6ST) to transduce its signal; as the IL-6 receptor lacks signaling properties, this signaling mode is often called IL-6 *cis-signaling*. This signaling cascade is only confined to specific cell types that express IL-6R on their surface, such as hepatocytes, some lymphocytes, and fibroblasts [6], and has been shown to procreate anti-inflammatory effects. However, in the case of inflammation, the concentration of proteases is considerably elevated, resulting in an increase in the soluble IL-6R (sIL-6R) concentration by proteolytic shedding [6]. The sIL-6R binds IL-6 with the same affinity as the membrane-bound receptor and thus forms a heterodimeric complex with sIL-6R in serum and transduces its signal through ubiquitously expressed GP130 [6,7]. This signaling mode of IL-6 is described as IL-6 *trans-signaling*, as it allows IL-6 to signal in cell types that lack IL-6R but express GP130 [8]. IL-6 *trans-signaling* has been shown to induce various pro-inflammatory signals in a range of diseases. Present therapeutic strategies, such as monoclonal antibodies targeting IL-6R (e.g., tocilizumab), have shown great potential in clinical trials of rheumatoid arthritis but are not optimal, as they target global IL-6 signaling and do not discriminate between *cis*- and *trans-signaling*; consequently, they can lead to undesirable side effects [8]. Therefore, therapeutic approaches targeting the pro-inflammatory *trans-signaling* of IL-6 would be highly beneficial. Similar to IL-6R, GP130 can also exist in a soluble form and can compete with the full-length isoform, thus blocking IL-6 *trans-signaling*. The generation of soluble GP130 endogenously is primarily regulated by alternative RNA processing (splicing or polyadenylation), with little evidence of proteolytic shedding [9]. These alternative splicing isoforms can antagonize IL-6 *trans-signaling* without affecting IL-6 *cis-signaling*. For instance, skipping exon 9 or 15 can lead to a premature stop codon, thus generating soluble isoforms that lack the transmembrane domain but retain the binding domains for the IL-6/sIL-6R heterodimeric complex [10,11,12,13]. Furthermore, exon 9 skipped isoforms have been shown to bind with IL-6/sIL-6R efficiently and have proven to be effective in controlling inflammation in mouse models of RA [10,11,12,13]. However, the levels of endogenous splicing events are insufficient to have a substantial therapeutic effect in acute and chronic inflammatory diseases. Therefore, strategies to modulate the endogenous splicing of *GP130* will be beneficial in countering IL-6 *trans-signaling*-mediated systemic inflammation.

Splice switching oligonucleotides (SSO) have proven successful in modulating RNA splicing for various clinical applications. SSOs are antisense oligonucleotides (AONs) targeted to bind splicing regulatory elements in pre-mRNA transcripts. The binding of SSOs to these elements modulates the binding of spliceosome components and other splicing factors involved in regulating alternative splicing, leading to exon skipping or exon inclusion [14]. SSOs have been widely used to redirect splicing in diseases such as ß-thalassemia, Duchenne muscular dystrophy, and spinal muscular atrophy [15]. The current field of AONs includes a broad spectrum of different RNA analogs with varying properties, including charged chemistries that are usually based on a phosphorothioate backbone (PS) in combination with ribose modifications such as 2′-O-methyl (2′-O-Me), locked nucleic acid (LNA), or 2′-O-methoxy-ethyl (2′-MOE), as well as charge-neutral chemistries such as peptide nucleic acid (PNA) or phosphorodiamidate morpholino oligomers (PMO) [14]. Importantly, all of these modifications prevent endo- and exonuclease cleavage, which in turn enhances the stability and bioavailability of SSOs.

Here, we employed SSOs to modulate the alternative splicing of *GP130* to generate soluble GP130 isoforms to precisely target IL-6 *trans-signaling*. Moreover, modulating the alternative splicing of *GP130* to generate soluble GP130 simultaneously enables the downregulation of the membrane-bound GP130, further reducing IL-6-mediated *trans-signaling*. We utilized state-of-the-art CPP-PMO conjugates for the efficient systemic in vivo splice switching of *Gp130*. Furthermore, we show the therapeutic potential of this novel approach in two different inflammatory animal models, where treatment with a Gp130 SSO improved therapeutic outcomes in the sepsis and Crohn’s disease models.

## 2. Materials and Methods

### 2.1. Cell Culture

Huh7 (human hepatocellular carcinoma cell line), HEK293T (human embryonic kidney cell line), HeLa (human cervical carcinoma cell line), Caco-2 (human colon adenocarcinoma cell line), N2a (mouse neuroblastoma cell line), the HeLa STAT3 luciferase reporter cell line (Signosis, Santa Clara, CA, USA), and the HEK-Blue™ IL-6 reporter cell line (InvivoGen, San Diego, CA, USA) were maintained in Dulbecco’s Modified Eagle’s Medium (DMEM) + GlutaMax™ (Gibco by Life Technologies, Waltham, MA, USA), supplemented with 10% heat-inactivated fetal bovine serum (FBS) (Gibco by Life Technologies, Billings, MT, USA) and 1% penicillin (100 U/mL) and streptomycin (100 μg/mL) (Sigma Life Science, Darmstadt, Germany) at 37 °C, in a 5% CO_2_ atmosphere. C2C12 (mouse myoblast cell line) cells were cultured in DMEM supplemented with 20% heat-inactivated FBS and 1% penicillin (100 U/mL) and streptomycin (100 μg/mL) at 37 °C, in a 5% CO_2_ atmosphere. Cells were passaged using 0.05% Trypsin–EDTA (1×) (Gibco by Life Technologies, Billings, MT, USA) when 80–90% confluency was reached.

### 2.2. Design of Splice Switching Oligonucelotides

When designing SSOs, there are many parameters that need to be taken into consideration in order to provide efficient SSOs. It is important to reflect on the length, thermodynamic properties, sequence content, and secondary structure of SSOs. Firstly, 3′ and 5′ splice sites were identified from existing EST data, and ESEfinder was used to determine the location of the exonic splicing enhancers. Secondly, the mFOLD server was used to imitate different secondary structures of pre-mRNA, which helped to appraise the accessibility of the target sequence in the pre-mRNA. SSOs are generally only 17 to 25 nucleotides long and previous studies have demonstrated that as such they are unlikely to form stable secondary structures. It was shown that by increasing the length of SSOs, the affinity to the target sequence should also increase, followed by increased splice switching activity. Moreover, longer SSOs have naturally higher overall Tm and their thermodynamic stability is higher. Regarding guanine and cytosine, it is suggested to keep the guanine–cytosine (GC) content low and avoid three or more stretches of these two bases as it is less likely for SSOs designed in this way to self-dimerize [16,17]. Considering all the guidelines, our 2′O-Me-PS were designed having 40% to 60% GC content, a melting temperature higher than 48 °C, and a length of 20 to 22 bp. SSO designs were run through BLAST to specify possible off-target binging. Lastly, selected SSO designs were synthesized by integrated DNA technologies. For Pip6a-PMO conjugates, Pip6a Ac-(RXRRBRRXRYQFLIRXRBRXRB)-COOH was synthesized and conjugated to PMO as described previously [18,19]. The PMO sequence was purchased from Gene Tools LLC, Philomath, OR, USA.
**Oligo Name****Sequence****Binding Site**hIL6ST-Ex9-1 (2′-O-Me +PS)5′ GCTTTAGATGGTCCTAAAGAAA 3′Exon 9 3′sshIL6ST-Ex9-2 (2′-O-Me +PS)5′ GGATGGATCTATTTTATACCAG 3′exon 9 ESEhIL6ST-Ex9-3 (2′-O-Me +PS)5′ ACCTTCCACACGAGTTGTAC 3′Exon 9 5′ssm/hIL6ST-Ex15-1 (2′-O-Me +PS)5′ ATTTCTCCTTGAGCTAAAACAA 3′Exon 15 3′ssm/hIL6ST-Ex15-2 (2′-O-Me +PS)5′ AUGGCUUCUAUUUCUCCUUGAGCU 3′Exon 15 3′ssm/hIL6ST-Ex15-3 (2′-O-Me +PS)5′ ACUUACAGGUCUCGUUUGUUAA 3′Exon 15 5′ssmIL6ST-Ex9-1 (2′-O-Me +PS)5′ GGTCTGGATGGTCCTAAAGAAA 3′Exon 9 3′ssmIL6ST-Ex9-2 (2′-O-Me +PS)5′ GGATGGATTTGTCTTATACCAG 3′Exon 9 ESEmIL6ST-Ex9-3 (2′-O-Me +PS)5′ ACCTTCCATATGAGCCGTAC 3′Exon 9 5′ssmIL6ST-Ex9-1 (PMO)5′ GGTCTGGATGGTCCTAAAGAAAAC 3′Exon 9 3′ssmIL6ST-Ex9-3 (PMO)5′ TTACCTTCCATATGAGCCGTAC 3′Exon 9 5′ss

### 2.3. SSO Transfection

Cells were transfected with SSOs 24 h after seeding. Transfections were performed using Lipofectamine^®^ 2000 (LF2000) (Invitrogen by Thermo Fisher Scientific, Waltham, MA, USA) as a transfection reagent and Opti-MEM™ I (Gibco by Life Technologies) as a complexation buffer. For the precomplexing of SSOs and LF2000 in a 1:1 ratio, they were incubated separately in Opti-MEM for 10 min. After this, LF2000 was added to SSOs and incubated for another 30 min at room temperature, followed by treating the cells with complexes. Cells were harvested 24 to 48 h post-transfection, depending on the exact experiment.

For the gymnosis (naked uptake) experiments, N2a and C2C12 cells were seeded in a 24-well plate at a cell density of 50,000 cells per well. The cells were maintained in the culture for 24 h, followed by treatment with PMO-modified SSOs (10 µM and 50 µM) for gymnosis. The cells were maintained in the licensed media for 72 h, followed by harvesting for RT-PCR analysis.

### 2.4. RNA Extraction and RT-PCR

For RT-PCR, RNA was extracted from the cells according to the manufacturer’s instructions using TRI^®^ reagent (Sigma-Aldrich, St. Louis, MI, USA). Once TRI^®^ reagent was added, samples were allowed to stand at room temperature for 5 min in order to ensure complete separation of nucleoprotein complexes. Chloroform (Sigma-Aldrich) was added and tubes were vigorously shaken for 15 s, followed by 5 min incubation at room temperature and 15 min centrifugation at 15,000× *g* and 4 °C. After this, the aqueous phase was then transferred into a new tube and 1 sup volume of isopropanol (Sigma-Aldrich) was added. Samples were vortexed and left for 10 min at room temperature, followed by 30 min centrifugation at 15,000× *g* and 4 °C. After the supernatant was removed, samples were washed twice with 75% EtOH and centrifuged for 10 min at 15,000× *g* and 4 °C. The pellet was left to air dry and dissolved in 20 μL previously warmed Milli-Q water. The concentration of RNA was determined with a NanoPhotometer NP80 (Implen, Munich, Germany). Then, 1 μg of RNA was reverse-transcribed with the High-Capacity cDNA Reverse Transcription Kit (Applied Biosystems by Thermo Fisher Scientific, Waltham, MA, USA) using MultiScribe™ Reverse Transcriptase (50 U/μL) in the presence of 10× RT Buffer, 10× RT random primers, 25× dNTPs (100 mM), and an appropriate volume of nuclease-free water, according to the manufacturer’s protocol. Reactions were incubated at 25 °C for 10 min, followed by 37 °C for 120 min and 85 °C for 5 min. PCRs were carried out with the HotStarTaq Plus Master Mix Kit (Qiagen, Hilden, Germany) using HotStarTaq Plus DNA Polymerase (5 U/μL) and the custom-designed primers listed below, purchased from Sigma-Aldrich, St. Louis, MO, USA (human and mouse Gp130 and GAPDH primers). Gp130 amplifications were carried out using Gp130 primers and PCR conditions starting with incubation at 95 °C for 5 min following 28 cycles of 95 °C for 30 s, 51 °C or 52.6 °C for 30 s, and 72 °C for 1 min, completing with a final extension at 72 °C for 10 min. GAPDH control cycling conditions for amplification were 95 °C for 5 min followed by 24 cycles of 95 °C for 30 s, 30 s at 51 °C for human GAPDH primers and 56 °C for mouse GAPDH primers, and 72 °C for 30 s. For the final extension, 72 °C for 10 min was used.

The success of PCR amplification was validated by gel electrophoresis using 2% agarose (Lonza, Basel, Switzerland) gel in 1× TAE buffer, stained with 0.625% ethidium bromide (VWR). Electrophoresis was run for 1.5 h on 70 V and 200 mA. Gels were then imaged using the Molecular Imager^®^ VersaDoc™ MP imaging system (Bio-Rad, Hercules, CA, USA).
**Primers****Sequence**hGp130 ex75′ AGGACCAAAGATGCCTCAACT 3′hGp130 ex105′ CCATCTTGTGAGAGTCACTTCATAATC 3′hGp130 ex145′ GCAGCATACACAGATGAAGGTC 3′hGp130 ex165′ TGGAGGAGTGTGAGGTGAC 3′mGp130 ex75′ AGGACCAAAGATGCCTCAACT3′mGp130 ex105′ TGACTGCGTAAGAATCACTTCATAATC 3′mGp130 ex145′ TGGTTGTGCATGTGGATTCT 3′mGp130 ex165′ TGGGCAATATGACTCTTGGA 3′h/m Gapdh F5′ ACATCGCTCAGACACCATG 3′h/m Gapdh R5′ TGTAGTTGAGGTCAATGAAGGG 3′

### 2.5. Luciferase Assay

For the experiment using SSOs, N2a cells were seeded in a 24-well plate (Corning Incorporated, Corning, NY, USA) at a density of 30,000 cells per well and HeLa STAT3 reporter cells in a 96-well plate (Corning Incorporated) at a density of 10,000 cells per well. N2a cells were maintained in the culture for 24 h and then treated with SSOs in a concentration of 50 nM and 100 nM. Cells were maintained in the culture for another 24 h before condition media were collected, spun down at 300× *g* for 5 min, and mixed with Hyper IL-6 (HIL-6), which was a kind gift from Stefan Rose-John, at a concentration of 2.5 ng per 500 μL of media. They were then transferred to HeLa STAT3 cells from which old condition media were previously removed. HeLa STAT3 cells were incubated with media for 8 h. After this, cells were lysed for 30 min using 0.1% Triton X-100 (VWR) and 25 μL of the lysate from each sample was transferred to a white 96-well assay plate with a flat bottom (Corning Incorporated, Corning, NY, USA). Luciferase was measured using Luciferase Assay Substrate (Promega, Madison, WI, USA) and a Glomax^®^ 96 microplate luminometer (Promega).

For experiments using TNF-α (Nordic BioSite, Täby, Sweden), the soluble IL-6 receptor (SIL-6R) (Sigma), and HIL-6, HeLa STAT3 cells were seeded in a 96-well plate at a density of 20,000 cells per well and maintained in the culture for 24 h, followed by treatment with 5 ng/mL of TNFα. At 4 h post TNF-α induction, HIL-6 and SIL-6 were added to the cells in concentrations of 5 ng/mL and 20 ng/mL, respectively, and incubated for 6 h. Afterwards, cells were lysed, and luciferase was measured as previously described. For the normalization of the luciferase readings, the DC™ Protein Assay (Bio-Rad, Hercules, CA, USA) was used following the manufacturer’s instructions and measured with the SpectraMax^®^ i3x reader (Molecular Devices, San Jose, CA, USA).

### 2.6. Western Blot

Cells were harvested using a cell scraper in ice-cold PBS, followed by centrifugation at 300× *g* for 5 min to spin down the cells. The supernatant was discarded and the cell pellet was dissolved in 60 µL of radioimmunoprecipitation assay buffer (RIPA) for lysis; lysed cells were incubated for 30 min on ice and vortexed every 5 min. After lysis, the lysate was centrifuged at 12,000× *g* for 10 min, in order to remove lipids. The supernatant was collected and dissolved in sample buffer (0.5 M dithiothreitol (DTT), 0.4 M sodium carbonate (Na_2_CO_3_), 8% sodium dodecyl sulphate (SDS), and 10% glycerol). The sample was then incubated at 65 °C for 5 min. Processed samples were then loaded onto a NuPAGE^®^ Novex^®^ 4–12% Bis-Tris Gel (Invitrogen, Life Technologies, Waltham, MA, USA) and run at 120 V in running buffer. The proteins on gel were then transferred to an iBlot nitrocellulose membrane (Invitrogen, Life Technologies, Waltham, MA, USA) using the iBlot system at 25V for 7 min. The membranes were subsequently blocked with Odyssey blocking buffer (LiCor, Lincoln, NE, USA) for 60 min at room temperature with gentle shaking. Followed by blocking, membranes were washed 5 times for 5 min each with PBS with 0.1% Tween-20 (PBS-T). After washing, membranes were incubated with primary antibody solution anti-Gp130 (Abcam, Cambridge, UK) and anti-STAT3 Try705 (Bioss, Woburn, MA, USA) overnight at 4 °C. Membranes were then washed 5 times for 5 min each with PBS-T. After washing, membranes were incubated with secondary antibody solution with anti-goat IgG DyLight-700 at 1:15,000 dilution for anti-Gp130 in a 1:1 ratio of Odyssey blocking buffer and PBS-T and anti-rabbit IgG DyLight-800 in a 1:1 ratio of Odyssey blocking buffer and PBS-T with 0.01% of SDS. After the secondary antibody incubation, membranes were washed 5 times for 5 min with PBS-T. Finally, the membrane was incubated in PBS and visualized on the LI-COR Odyssey CLX infrared imaging system.

### 2.7. Flow Cytometry

N2a cells were plated in a 24-well plate at a density of 75,000 cells per well and kept in the culture for 24 h, followed by 100 nM SSO transfections. After 24 h of incubation, the condition media were removed and cells were trypsinized. The reaction was stopped with 200 μL of fresh DMEM, from which 100 μL was then transferred to a new 96-well plate with a V-shaped bottom (Corning Incorporated, Corning, NY, USA). Then, 3 μL of rat IgG anti-mouse Gp130 allophycocyanin (APC)-conjugated antibody (R&D Systems, Minneapolis, MN, USA) was added to half of the samples, followed by 30 min incubation at 4 °C. The other half of the samples served as a control and nothing was added to them. Following incubation, 100 μL of PBS was added and the plate was centrifuged for 5 min at 900× *g*. The supernatant was discharged and 100 μL of PBS containing 10× 4′,6-diamidine-2′-phenylindole dihydrochloride (DAPI) (Sigma-Aldrich, St. Louis, MO, USA) was added to all the samples. The plate was then analyzed using a MACSQuant^®^ Analyzer 10 (MACS Miltenyi Biotec, Teterow, Germany) flow cytometer and data were processed with the FlowJo^®^ analysis platform (FlowJo, LLC, Vancouver, BC, Canada). Three gates were applied to investigate and quantify the populations of interest. With the first gate, viable cells were selected; then, gating for single cells was used and lastly Gp130-positive cells were chosen.

### 2.8. Systemic Inflammation Model

Systemic inflammation was induced using LPS as described previously [20]. Female C57BL/6 mice (20 ± 5 g) were injected i.p. with LPS (L-5886, Sigma-Aldrich, St. Louis, MO, USA). Pip6a-PMO was i.v. injected through the tail vein 24 h before LPS induction and the animals were observed and weighed every 4 h after induction. IL-6 and TNF-alpha levels were assessed in blood plasma using ELISA (biolegend, San Diego, CA, USA) based on the manufacturer’s protocol. To assess the in vivo efficacy of Pip6a-PMO, conjugates were intravenously injected through the tail vein in wild-type female NMRI mice (20 ± 5 g). Then, 72 h post-injection, each animal was euthanized and different organs were harvested for RT-PCR analysis. The animal experiments were approved by the Swedish Local Board for Laboratory Animals.

### 2.9. TNBS-Induced Colitis Model

TNBS-induced colitis was induced as described previously [20]. Female BALB/c mice (20 ± 5 g) were presensitized with the peritoneum skin application of 60 µL 5% TNBS + 90 µL acetone–olive oil (4:1) mix per mouse. One week later, colitis was induced by the intrarectal administration of 30 µL TNBS + 42.1 µL 95% ethanol + 27.9 µL H_2_O per mouse. Mice were subsequently monitored for changes in body weight. For therapeutic treatment, Pip6a PMO was injected intravenously a day before induction as a single i.v. injection. The animal experiments were approved by the Swedish Local Board for Laboratory Animals.

## 3. Results

### 3.1. Design of Splice Switching Oligonucleotides

GP130 is a transmembrane protein with three N-terminal Ig-like domains followed by three fibronectin-type III domains. Until now, three different functional sGP130 isoforms with molecular weights of 50, 90, and 110 kDa have been detected in human serum. Importantly, all of these isoforms are generated primarily by alternative splicing or polyadenylation, as limited proteolytic cleavage has been reported to date (Figure 1). The smallest known *GP130* isoform that retains binding to the IL-6/sIL-6R isoform is GP130 RAPS [9]; here, exon 9 exclusion generates a premature stop codon in exon 10 by a shift in the open reading frame. The resultant 50 kDa isoform comprises 1–3 IgG-like domains and can efficiently downregulate IL-6 *trans-signaling*. Importantly, none of the tested human or mouse cells used in this study showed any exon 9 skipped *GP130* isoform in RT-PCR (Appendix A), therefore making it an ideal candidate to evaluate the efficacy of SSOs, as the background exon 9 splicing event is minimal and allows for the easy assessment of splice switching.

To induce exon 9 skipping in *GP130*, we designed three different SSOs targeting the intron 8 3′ splice site (3′ss), exon 9 splicing enhancer, and intron 9 5′ splice site (5′ss) (Figure 2A). The SSOs were designed based on previously published parameters [17] with 2′-O′Me modification on the sugar and phosphorothioate interlinkages to enhance the SSOs’ stability and binding efficiency against their respective targets. We observed robust splice switching in a range of human cell lines upon transfection of the SSOs (Figure 2B and Appendix A). SSOs targeting the intron 9 5′ss showed robust splice switching in a dose-dependent manner. Notably, the alternative splicing pattern of *GP130* is also conserved in other species. Therefore, for a pre-clinical evaluation of this approach in mice, we designed and evaluated SSOs for the induction of exon 9 skipping in mouse *Gp130*. We observed a similar outcome with the mouse GP130 SSOs, where targeting the 5′ss showed robust exon skipping (Figure 2C,D).

In addition to the exon 9 skipped isoform, a longer soluble isoform can also be generated by alternative splicing, such as exon 15 skipping. Exon 15 skipping also leads to a premature stop codon and drives the expression of a 110 kDa soluble GP130 protein with a similar binding affinity as the exon 9 skipped soluble GP130 isoform (Figure 1) [9]. Importantly, inducing a larger sGP130 may offer better pharmacokinetic properties than the exon 9 skipped isoform. To test the feasibility of this approach, we designed SSOs targeting the splice sites of exon 15 of both human and mouse pre-mRNA, respectively. We observed similar splice switching efficacy both in mouse and human cells, with 100 nM SSO oligo transfection leading to up to 50% splice switching (Figure 3A–D).

### 3.2. Exon Skipped Isoform Downregulates IL-6 Trans-Signaling

With robust splice switching observed with both the GP130 exon 9 and exon 15 SSOs, we next evaluated the effect of splice switching on IL-6 *trans-signaling*. For the quantitative assessment of IL-6 signaling, we used a reporter HeLa cell line engineered with IL-6R and a STAT3-driven luciferase gene expression cassette. This cell model serves as an excellent tool to study IL-6 *cis-* and *trans-signaling*, as the addition of hyper IL-6 (a fusion protein of IL-6 and sIL-6 R that mimics the *trans-signaling* complex [21]) induced STAT3 phosphorylation and the subsequent gene expression of firefly luciferase (Appendix A). To assess the effect of splice switching on IL-6 *trans-signaling*, we used a trans-complementation assay, where conditioned medium (48 h) from cells transfected with SSOs 48 h prior was added to reporter cells along with hyper IL-6. We observed the statistically significant downregulation of IL-6 *trans-signaling* with the conditioned medium from cells transfected with an SSO targeting exon 9 (Figure 4A). Furthermore, in HeLa cells, treatment with IL-6 + sIL-6R along with conditioned medium from cells transfected with exon 9 SSOs showed the downregulation of STAT3 phosphorylation as compared to the control oligo group (Figure 4B). Furthermore, cells transfected with SSOs showed the downregulation of the cell surface GP130 receptor as determined by flow cytometry (Figure 4C). These data collectively indicate that SSOs potently convert the membrane-bound GP130 isoform to a soluble GP130 isoform that is capable of downregulating IL-6 *trans-signaling*.

### 3.3. In Vivo Splice Switching of Gp130 with Pip6a PMO

The cellular uptake and bioavailability of nucleic acid drugs such as SSOs is always a matter of concern, as the naked uptake of SSOs requires high doses, and there is a lack of tissue specificity or cell type targeting [15]. In addition, systemic toxicity associated with a high dose of SSOs and disease severity can strongly exacerbate the clinical outcome. Therefore, we opted for PMO-modified SSOs, a safer and clinically used AON chemistry, to address these challenges. Notably, PMOs are charge-neutral, substantially reducing endogenous protein binding, resulting in a safer toxicity profile [14]. However, due to the lack of endogenous protein binding, the bioavailability of PMO SSOs is severely compromised; therefore, higher doses are often required to observe therapeutic effects [22]. Similar observations were also made in our experiments, where exon 9 targeting PMO SSOs showed a modest effect at doses as high as 50 µM (Appendix A). Therefore, using a non-viral vector to facilitate intracellular delivery is vital to unlocking the full potential of PMOs. One approach involves the direct conjugation of cell-penetrating peptides (CPPs) such as Pip6a to the PMO [18]. This allows for the efficient delivery of PMO SSOs to extrahepatic tissue such as the muscle, heart, and brain. The covalent conjugation of the Gp130 exon 9 PMO SSO to Pip6a enhanced the delivery in vitro, with up to 90% exon skipping observed at a 10 µM concentration (Figure 5A). Next, we evaluated the in vivo efficacy of Pip6a PMO conjugates by intravenous delivery in wild-type NMRI mice. We observed robust splice switching following a single injection of 15 mg/kg of Pip6a-PMO at 3 days in the heart, gastrocnemius muscle, liver, spleen, kidneys, and gastrointestinal tract (Figure 5B). In contrast, at a lower dose of Pip6a-PMO (5 mg/kg), modest splice switching in the liver only was detected. Importantly, the difference in activity in different tissues was likely due to differences in the in vivo uptake of Pip6a-PMO.

### 3.4. Therapeutic Applicability in Two Different Mouse Models

After establishing the in vivo potency of the conjugates, we next evaluated the therapeutic efficacy of these SSOs in two different systemic inflammation models where IL-6 *trans-signaling* has been shown to play an important role [23]. We first assessed the therapeutic applicability of *Gp130* exon 9 SSOs in an LPS-induced inflammation model in mice, which clinically mimics sepsis and where IL-6 plays a critical role in the pathophysiology. Intravenous administration of Pip6a-Gp130 exon 9 PMO at 15 mg/kg 24 h before LPS administration showed improved survival compared to the LPS only group (100% vs. 80%) (Figure 6A). In addition, IL-6 and TNF-alpha plasma levels at 24 h post disease induction were downregulated in animals that received SSOs compared to mice treated with LPS only (Figure 6B,C). Notably, the drop in IL-6 plasma levels in the SSO-treated group could also be due to the lack of sensitivity of the ELISA antibody to detect IL-6/sIL-6R/sGp130 tetrameric complexes and may require further investigation in the future. Next, we evaluated the efficacy of *Gp130* exon skipping in a TNBS-induced mouse colitis model, which clinically mimics Crohn’s disease in humans. The intravenous administration of Pip6a-Gp130 exon 9 PMO at 15 mg/kg 24 h prior to colitis induction showed a lower drop in body weight compared to saline-treated animals (Figure 6D). These observations clearly indicate the therapeutic potential of *Gp130* exon skipping to treat systemic inflammatory diseases.

## 4. Discussion

In this study, we describe a novel therapeutic modality against chronic inflammation using SSOs to generate a soluble GP130 isoform. Soluble GP130 (sGP130) is a naturally occurring glycoprotein that specifically inhibits IL-6 *trans-signaling* through the soluble IL-6 receptor (IL-6R), without affecting the classic signaling pathway mediated by the membrane-bound IL-6R. IL-6 signaling plays a crucial role in inflammation and cancer development, and several monoclonal antibodies targeting IL-6 or IL-6R have been developed as therapeutic agents for inflammatory diseases and cancers [24]. For instance, tocilizumab, targeting the IL-6R, is approved for rheumatoid arthritis, juvenile idiopathic arthritis, and Castleman disease, while sarilumab, also targeting the IL-6R, is in late-stage clinical trials [8]. However, it is important to note that a global blockade of IL-6 signaling pathways can affect both pro- and anti-inflammatory properties, potentially adversely affecting the immune system. Careful consideration and understanding the balance between inflammation control and immune regulation are essential in developing IL-6-targeted therapies [8]. Based on this, soluble GP130 proteins have been repurposed for the therapeutic blockade of IL-6 *trans-signaling* and have shown promising results in a range of pre-clinical inflammatory disease models [20,24,25,26,27]. Importantly, protein biotherapeutics often display poor pharmacokinetics due to renal clearance, and periodic repeated administration is needed to maintain the therapeutic efficacy. Therefore, a nucleic acid-based approach, as described here, holds the potential to provide a long-lasting effect as these SSOs are chemically modified for prolonged stability inside the cells and act as a catalytic effector on splicing [14].

Our study used SSOs modified with 2′O-Me and phosphorothioate (PS) for initial screening. This modification set is one of the most widely used AON chemistries for SSOs and other antisense oligonucleotide applications. Overall, these SSOs confer nuclease resistance and enhance on-target binding. In addition, the phosphorothioate backbone modification enhances the cellular delivery of SSOs by interacting with various serum proteins to facilitate efficient endocytosis in the target cell [15]. In recent years, with advances in nucleic acid chemistry, a range of novel chemical modifications have become available, such as 2′MOE, LNA, or Tricyclo-DNA [28]. Although these modifications, in part, address the in vivo applicability of nucleic acid drugs by improving safety and on-target binding, the efficacy of SSOs is primarily dependent on the design, and, for in vitro design optimization, standard chemical modifications are more economical and easier to manufacture.

Regarding the design of SSOs, we followed the guidelines from previous studies describing SSOs designed for exon skipping in dystrophin [17]. We designed all SSOs targeting the splicing of the regulatory elements of the target exon predicted by the ESE splice finder algorithm, with GC content between 40 and 60% and 20–22 nucleotides in length. Both exon 9 skipped and exon 15 skipped sGP130 mRNAs were successfully obtained upon treatment with the designed SSOs targeting either the 5′ss or 3′ss. In this study, SSOs targeting exon–intron boundaries tended to be more potent than the ones directed towards ESEs. Importantly, mouse designs of SSOs tested in vitro in both N2A and C2C12 cells resulted in similar efficacy compared to the respective human counterparts.

Regarding functional activity, the SSO-induced exon skipped product showed the downregulation of IL-6 trans-signaling in a reporter assay. Notably, a slight decrease in soluble GP130 levels was observed in the conditioned medium with ELISA. This discrepancy could be due to the lack of affinity for shorter GP130 isoforms by the antibody used in ELISA. In addition, since the exon 9 skipped isoform is the smallest known isoform, inducing exon 9 skipping will result in a reduction in the level of all the other, longer, soluble and membrane-bound GP130 isoforms. In further support of this, we also demonstrated by flow cytometry that the SSO treatment strategy affects the entire GP130 receptor levels and not only the soluble form. Moreover, there is the possibility that this mRNA isoform has a shorter half-life, e.g., by being subject to nonsense-mediated decay. However, this isoform has been found naturally in rheumatoid arthritis patients. Furthermore, no increase in the splice-switched isoform was observed upon blocking nonsense-mediated decay by cycloheximide. This indicates that this splice-switched mRNA isoform is stable.

To translate this approach into a therapeutic application, efficient and safe in vivo delivery is key. Despite recent advances in nucleic acid chemistry, toxicity associated with an effective dose of charged nucleic acids can impose severe adverse effects in chronic inflammatory conditions. To mitigate the toxicity related to SSO delivery, we opted for CPP-PMO conjugates for efficient in vivo delivery. In recent years, CPP-PMO conjugates have emerged as a novel platform for potent in vivo delivery and have shown excellent efficacy in a range of pre-clinical models, including Duchene muscular dystrophy, spinal muscular atrophy, and myotonic dystrophy for the delivery of SSOs [29,30]. Here, we utilized Pip6a as our lead CPP, as it has been previously shown to deliver SSOs to the muscle, brain, and heart. Our study observed potent splice switching in the liver and other tissues at a 15 mg/kg dose. Importantly, our approach aims to generate a soluble GP130 protein isoform, and since GP130 is ubiquitously expressed in all the tissues, the tissue-specific delivery of SSOs is not vital for the therapeutic approach. This was evident with a phenotypic improvement in two different chronic inflammatory disease models, where prophylactic treatment improved survival in LPS-induced systemic inflammation and weight recovery in a TNBS-induced colitis model. Importantly, the modest in vivo activity could be due to the poor bioavailability of SSOs in vivo. In addition, SSO design optimization may improve the in vivo efficacy of GP130 SSO in the future. In conclusion, this novel approach offers broader therapeutic applicability for various inflammatory diseases by explicitly targeting the pro-inflammatory IL-6 *trans-signaling*.

## Figures and Tables

**Figure 1 cells-12-02285-f001:**
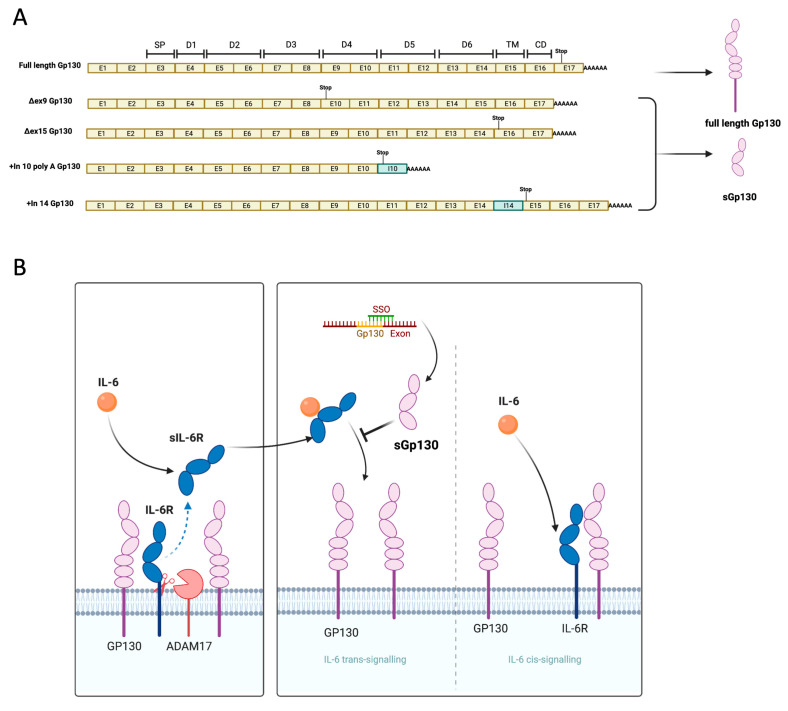
(**A**) Brief overview of the Gp130 alternative splicing and known protein isoforms. (**B**) SSO mediated sGp130 production to inhibit IL-6 *trans-signaling* specifically.

**Figure 2 cells-12-02285-f002:**
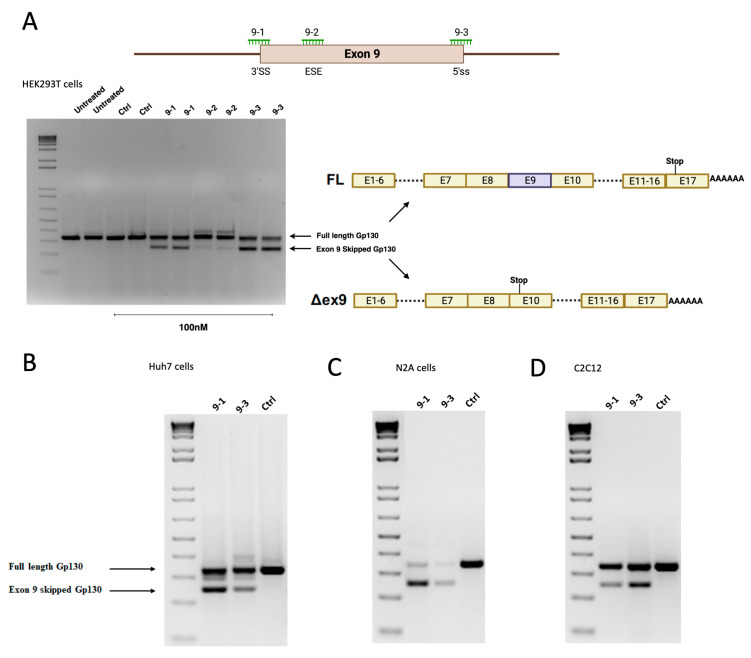
(**A**) Design and activity of different SSOs targeting Gp130 exon 9 splicing regulatory elements. End-point RT-PCR of Gp130 exon 9 skipping in HEK293T cells 24 h post-transfection of 100 nM Gp130 exon 9 SSOs by lipofectamine 2000. Splice switching efficiency of different Gp130 SSOs in (**B**) Huh7 cells, (**C**) mouse N2a cells, and (**D**) mouse C2C12 cells 24 h after transfection. The SSOs for mouse Gp130 were designed to bind at the positions corresponding to the those of the human Gp130 SSOs.

**Figure 3 cells-12-02285-f003:**
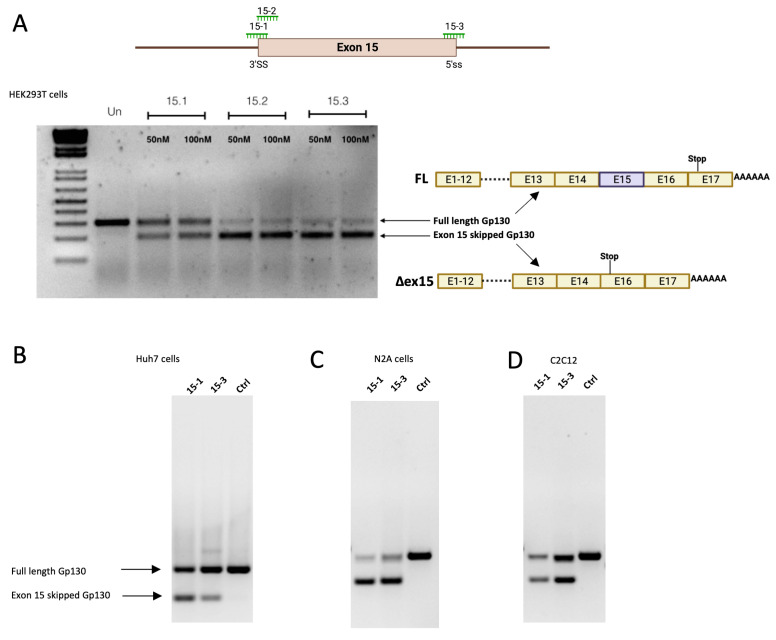
(**A**) Design and activity of different SSOs targeting Gp130 exon 15 splicing regulatory elements. End-point RT-PCR of Gp130 exon 15 skipping in HEK293T cells 24 h post-transfection of 100 nM Gp130 exon 15 SSOs by lipofectamine 2000. Splice switching efficiency of different Gp130 SSOs in (**B**) Huh7 cells, (**C**) mouse N2a cells, and (**D**) mouse C2C12 cells 24 h after transfection. The SSOs for mouse Gp130 were designed to bind at the positions corresponding to the those of the human Gp130 SSOs.

**Figure 4 cells-12-02285-f004:**
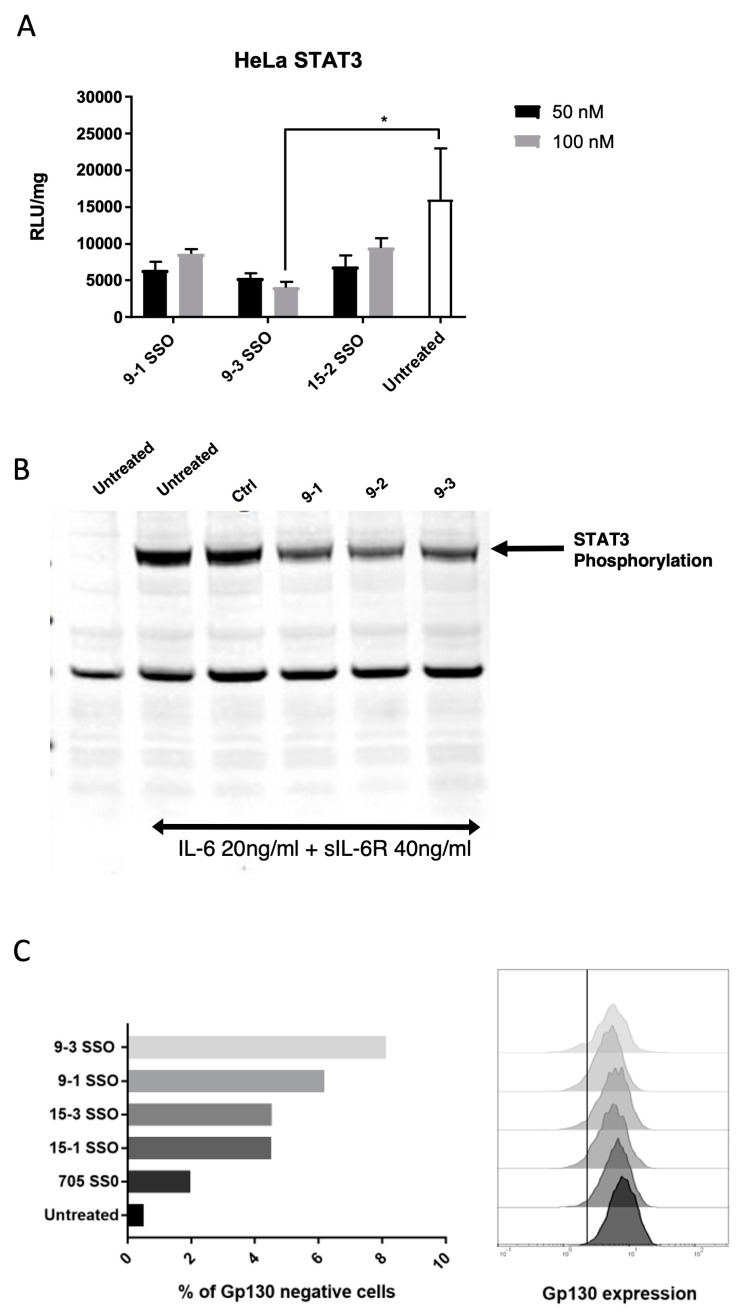
SSO-induced soluble Gp130 downregulates IL-6 trans-signaling in HeLa cells. (**A**) RLU/mg in HeLa STAT3 Luc cell lysate 24 h after the addition of 10ng/mL hyper IL-6 and conditioned medium from N2A cells transfected with Gp130 SSOs. Statistical analysis was done with two-way ANOVA and Fischer’s LSD test. * = *p* < 0.05. Data represent mean + SD of three replicates. Only 100 nM 9-3 SSO induced a significant effect compared with untreated. (**B**) STAT3 phosphorylation as determined by Western blot in HeLa cells 6 h post-treatment with IL-6 + sIL-6R cocktail and conditioned medium from HEK293T cells transfected with Gp130 SSOs. (**C**) Gp130 exon skipping downregulates full-length Gp130 isoform in mouse N2A cells. Flow cytometry analysis of the mGp130 expression after the treatment with 9-1, 9-3, 15-1, and 15-3 SSOs, and 705 SSO as a negative control.

**Figure 5 cells-12-02285-f005:**
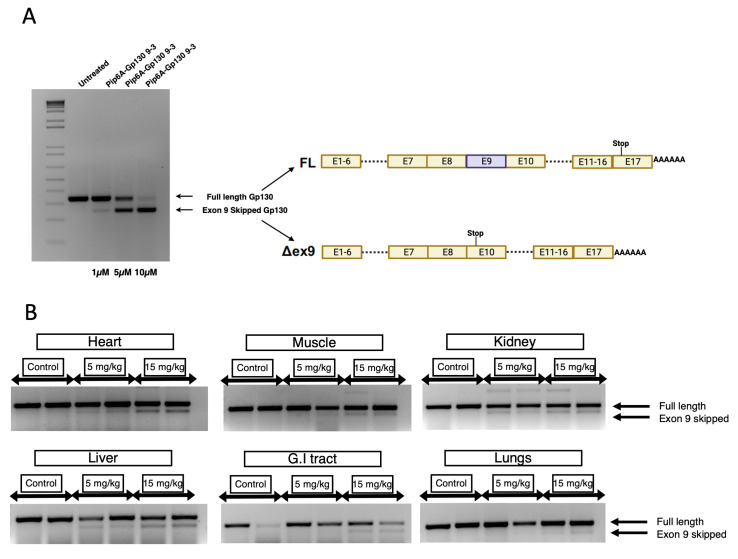
(**A**) Covalent conjugation of PMO with Pip6a CPP enhances splice switching in vitro and in vivo. End-point RT-PCR of Gp130 exon 9 skipping in mouse N2A cells treated with varying amounts of Pip6a-Exon 9 PMO SSO. (**B**) Pip6a-PMO conjugates show efficient in vivo exon skipping upon systemic delivery in wild-type mice. End-point RT-PCR of Gp130 exon 9 skipping in different tissues 72 h post intravenous administration of Pip6a PMO at 5 mg/kg or 15 mg/kg.

**Figure 6 cells-12-02285-f006:**
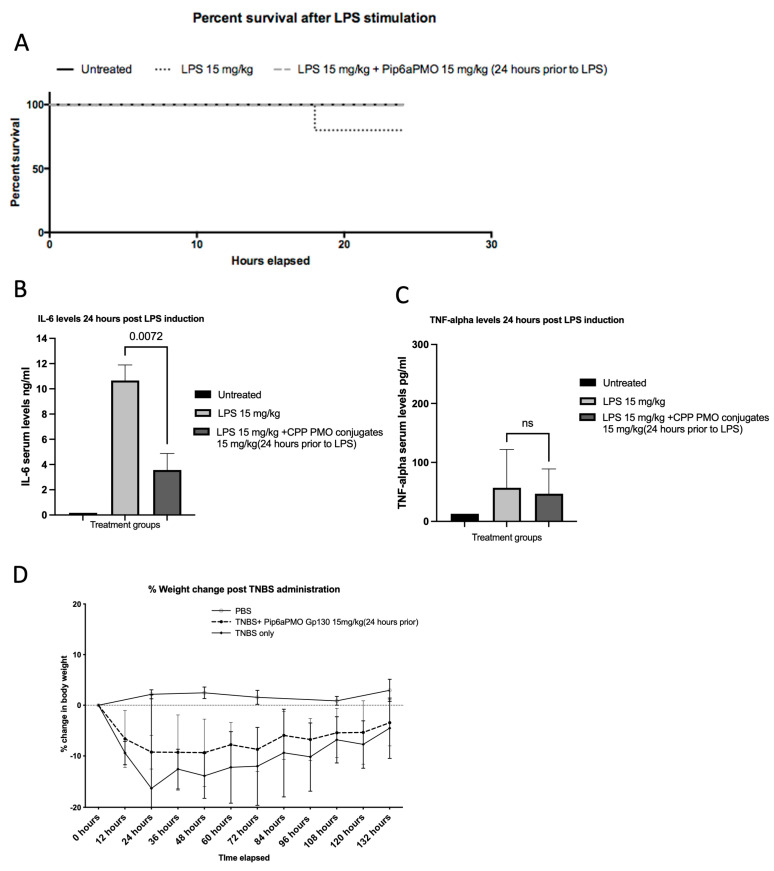
(**A**) Gp130 exon 9 SSOs improve survival and disease phenotype in LPS-induced systemic inflammation. C57Bl6J were injected intravenously either with 15 mg/kg Pip6a-Gp130 PMO or saline, and, 24 h later, systemic inflammation was induced by intraperitoneal administration of 15 mg/kg LPS (n = 5). (**A**) Kaplan–Meier survival curve of animals with and without SSO treatment post LPS induction. Plasma levels of (**B**) IL-6 and (**C**) TNF-alpha as determined by ELISA 24 h post disease induction. Significance calculated by one-way ANOVA. (**D**) The percentage change in body weight relative to the initial weight over the disease course of mice induced with colitis by intrarectal injection of TNBS and treated intravenously 24 prior with either 15 mg/kg Pip6a-PMO or saline. Significance calculated by two-way ANOVA, data non-significant.

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
