# Peer review of "Modulation of Pro-Inflammatory IL-6 Trans-Signaling Axis by Splice Switching Oligonucleotides as a Therapeutic Modality in Inflammation"

_cells, 2023, doi:10.3390/cells12182285_

Round 1

Reviewer 1 Report

The study presented in the manuscript addresses a significant problem in IL6 research - the ability to target trans signaling of IL6. The study will also have significant impact on development of future therapies against inflammatory diseases involving IL6.

While the study's objective is novel and impactful, there are scientific limitations and need to be addressed. A few of these limitations are given below:

1. Data show that SSOs highly efficient in inducing gp130 splicing in the cell lines that do not have any detectable native splicing of gp130. However, did the authors see similar effectiveness of SSOs in cells that exhibit native gp130 splicing - i.e., cells that already secrete soluble gp130?

2. How stable are these SSOs in cell culture medium and mouse serum? As they are required in high concentrations even in the presence of delivery system, are their half-lives very low? Any half-life measurements in cell culture medium or serum done by the authors? How effectively are the SSOs distributed in different organs in mouse body?

3. As mentioned above, the splicing effect seen in different mouse organs have varying efficacy. Are these differences because of variations in SSO uptake by these organs or variations in bioavailability of SSOs in these organs?

4. Do the SSOs, being nucleic acids, stimulate innate immune response by themselves? Did the author measure IFN levels in mice treated with just the SSOs?

4. One serious drawback in this study is the lack of statistical analysis, mainly in Figure 6. Were the in vivo survival differences, cytokine levels and body weight differences significantly significant? What was the p-value? If statistics were done, please add the p-value to the figures. Otherwise, the impact of the study will not be evident to the readers.

5. In Figure 6B, is the drop in IL6 levels only due to the anti-inflammatory response by soluble gp130? An alternative explanation could be that in the presence of soluble IL6R and spliced soluble gp130, the IL6 could be trapped in heterotrimeric complexes and could not be detected by the assay? Does the ELISA kit react with IL6 regardless of whether it is free or complexed with receptors? Any comments from the authors on this?

6. Authors claim that TNF is downregulated after SSO treatment in mice (Figure 6C). However, the difference is minor and within the standard error values. Representative figure with statistics is appreciated.

Reviewer 2 Report

1.     Authors should also validate the protein translation from alternative spliced gp130 mRNAs by western blots. 

2.     Authors should clarify that STAT3 driven luciferase assay is not specific to IL-6 tans-signaling as both cis- and trans-signaling share the downstream components. 

3.     In figure 4A, treatment of SSO in Hela cells showed minor effect in reducing IL-6-STAT3 signaling. Author should also check the efficiency of SS0 on GP130 alternative splicing by GP130 qPCR and western blot. The same as Figure4c, the differences of membrane GP130 expression in different treatment is very minor. Author should try STAT3 reporter assay with other cell lines which have been shown better response to SSO. 

4.     Statistic analysis is missing in figure6. In Figure6A, LPS treated group should also give control PMO at the same time point as Pip6aPMO group. Also, higher concentration of LPS can be used to have better idea of the protective effect of IL-6-stat3 signaling blockage. 

5.     Figure6D can hardly be statically significant to claim that Pip6aPMO treatment has protective effect on TNBS-mediated wight loss. 

6.     In general Figure5 and 6 showed very minor effect in vivo. 

Round 2

Reviewer 1 Report

Questions raised by the reviewer have been answered and relevant modifications have been made the text. No further questions to the authors.

Reviewer 2 Report

Most of concerns are addressed.